# Quantification of Squalene and Lactic Acid in Hair Bulbs with Damaged Sheaths: Are They Metabolic Wastes in Alopecia?

**DOI:** 10.3390/biomedicines11092493

**Published:** 2023-09-08

**Authors:** Diego Romano Perinelli, Alessandra Cambriani, Gianluigi Antognini, Gaetano Agostinacchio, Andrea Marliani, Marco Cespi, Elisabetta Torregiani, Giulia Bonacucina

**Affiliations:** 1Chemistry, Interdisciplinary Project (ChIP), School of Pharmacy, University of Camerino, Via Madonna delle Carceri, 62032 Camerino, Italy; diego.perinelli@unicam.it (D.R.P.); alessandra.cambriani@unicam.it (A.C.); giulia.bonacucina@unicam.it (G.B.); 2S.I.Tri.—Italian Society for Hair Science and Restoration, Via San Domenico 107, 50133 Florence, Italy; g_antognini@virgilio.it (G.A.); agostinacchio.g@libero.it (G.A.);

**Keywords:** high-pressure liquid chromatography, optical microscopy, solvent extraction, analytical quantification, hair loss, follicle

## Abstract

Alopecia is a pathological and multifactorial condition characterised by an altered hair growth cycle and ascribed to different pathogenic causes. Cell energetic imbalances in hair follicles occurring in this disorder could lead to the production of some “metabolic wastes”, including squalene and lactic acid, which could be involved in the clinically observed sheath damage. The aim of this work was the extraction and analytical quantification of squalene and lactic acid from hair bulbs of subjects with clinical alopecia in comparison with controls, using HPLC-DAD and HPLC-MS techniques. The analytical quantification was performed after a preliminary observation through a polarised optical microscope to assess sheath damage and morphological alterations in the cases group. A significantly larger amount of squalene was quantified only in subjects affected by alopecia (*n* = 31) and with evident damage to hair sheaths. For lactic acid, no statistically significant differences were found between cases (*n* = 21) and controls (*n* = 21) under the experimental conditions used. Therefore, the obtained results suggest that squalene can represent a metabolic and a pathogenic marker for some alopecia conditions.

## 1. Introduction

Alopecia is a pathological condition related to excessive hair loss from the scalp. It is a multifactorial disorder which can be ascribed to different pathogenic causes such as family history (hereditary), hormonal changes, medical conditions, and physical or psychological stress [1]; therefore, in the medical scenario, different forms of alopecia have been recognised such as androgenetic alopecia, alopecia areata, and others [2,3]. All forms of alopecia are characterised by an altered hair growth cycle. Hair growth is a continuous process characterised by four phases: anagen (growth), catagen (regression, i.e., transition from anagen to telogen, lasting about two weeks); telogen (rest, from two to three months), and exogen (shedding). Individual hair follicles independently undergo ten to thirty cycles in a lifetime, and for healthy subjects, the anagen-to-telogen ratio is approximately 14:1 to 12:1 [4]. On the other side, all subtypes of alopecia are characteristic of the premature entry of anagen follicles into telogen; therefore, the time from the anagen to telogen phase is shortened [5]. Different factors such as inflammation, hormones, stress, nutritional deficiency, poor sleep quality, and cellular-division-inhibiting medication can increase anagen-to-telogen transition, favouring hair loss [4,6,7,8]. Among them, stress is also one of the main causes, and diffuse hair thinning is often visible in subjects characterised by psychological stress [9,10]. Stress is commonly associated with the induction of inflammation, even in hair follicles, due to perifollicular macrophage clusters and excessive mast cell activation [6]. In addition to the activation of the hypothalamic–pituitary–adrenal cortex axis, there is an alternative, peripheral, and cutaneous pathway involved in the response to stress. This involves a chain of neuropeptides, neurotransmitters, and hormones, which are released from the cutaneous nerve plexuses, including nerve growth factor, substance P, and catecholamines, which are the key mediators of the inhibitory effects on stress-induced hair growth [11,12,13]. The sensory plexus, under the stimulus of nerve growth factor, releases substance P, which sustains inflammation [13]. The sympathetic plexus, under stress, is also able to release norepinephrine in the perifollicular intercellular spaces, which is a powerful vasoconstrictor and inhibitor of adenyl cyclase, which can inhibit the entire kinase cascade in the metabolic process (i.e., glycolysis, hexose monophosphate pathway, and Krebs cycle) [14]. When energy metabolism is blocked, the mitosis of the hair matrix can be inhibited. 

An excess of adrenergic tone (i.e., norepinephrine) in the follicular system, causing vasoconstriction, leads to ischemia and hypoxia, which promote glycolysis and the formation of lactic acid [15]. When the concentration of lactic acid is supraphysiological, the pH of the follicle is lowered, thereby inhibiting the Krebs cycle, so pyruvic acid cannot be metabolised with the accumulation of lactic acid. Cell metabolic imbalance can also have an impact on lipid metabolism. The metabolism of triglycerides is normally directed towards the formation of fatty acids but can also be diverted to acetyl coenzyme A, which, if not efficiently disposed of in the Krebs cycle, promotes the biosynthesis of sterols such as squalene, which could accumulate in hair [16,17]. In clinical practice, morphological alterations to hair have been visualised using polarised light optical microscopy, which could be related to the accumulation of “metabolic wastes”, such as lactic acid and squalene, leading to a macroscopic sheath damage. These morphological alterations have been observed with preclinical or clinical conditions that have been correlated with the development of alopecia [18]. Therefore, lactic acid and squalene are metabolic products that could be involved in the pathogenesis of hair loss since, once accumulated, they can damage hair sheaths, thereby altering hair growth. Despite all these assumptions, the presence of lactic acid and squalene in the bulbs of hair, displaying morphological alterations on sheaths, has never been assessed or analytically quantified. The aim of this work was the extraction and analytical quantification of squalene and lactic acid from hair bulbs of subjects selected on the basis of their clinical history of alopecia and after a preliminary evaluation of the bulbs through polarised light optical microscopy. The determined squalene and lactic acid values were then compared with those obtained from the group of controls, made up of apparently healthy subjects, in order to correlate the sheath damage observed using polarised light microscopy with the presence of these two compounds in the hair bulbs.

## 2. Materials and Methods

### 2.1. Materials

Sodium hydroxide, hydrochloric acid 37% aqueous solution, formic acid, acetonitrile, n-hexane, and methanol were purchased from Carlo Erba Reagents srl (Cornaredo, Italy). Lactic acid (Fluka, Segrate, Italy) and squalene (Supelco^®^, East Riding, UK) of analytical standard grade were purchased from Merk KGaA (Darmstadt, Germany).

### 2.2. Bulbs Sampling

Hair was extracted from scalp using haemostatic forceps from a point on the vertex of the head. The extraction point on the scalp was the point where an imaginary line drawn from the nose to the external occipital protuberance met a second line passing from ear to ear on the frontal. After extraction, hair samples were cut to 3 mm from the end to separate bulbs. The extraction procedure was conducted by dermatologists in a medical clinic as a part of their routine practice.

### 2.3. Bulbs Observation under Polarised Microscope

Bulbs were put on a glass slide and wetted with a drop of immersion oil for microscopy (refractive index 1.515–1.517, Titolchimica S.p.a, Pontecchio Polesine, Italy), covered with a coverslip, and then observed under a polarised light optical microscope (Optika B-500 series, Ponteranica, Italy, 4×/0.1 ocular magnification) equipped with a Sony A6000 camera. The same hair samples observed through the microscope were utilised for the extraction of squalene and lactic acid after removing the excess of immersion oil by blotting.

### 2.4. Inclusion Criteria

Subjects for the study were chosen according to the following inclusion criteria: men and women (aged 30–60 years) who presented a clinical picture compatible with androgenetic alopecia as evaluated by medical anamnesis, trichoscopy, and the presence of miniaturised hair. All subjects showed evident damage to the sheaths at different extents observed by polarised optical microscopy. These subjects were grouped and referred as “cases”. The controls were selected among subjects with no clinical picture compatible with androgenetic alopecia and no damage to the sheaths. 

### 2.5. Squalene and Lactic Acid Extraction from Bulbs

The bulb was first incubated in 4 mL of 0.5 M NaOH at 80 °C for 90 min under reflux. After cooling down, the sample was neutralised with HCl 0.1 M, and the solution was transferred to a separating funnel. Then, five consecutive extractions were performed using 5 mL of hexane for each bulb. The aqueous and organic phases were collected separately, containing lactic acid and squalene, respectively. The hexane collected from the 5 consecutive extractions was then evaporated under vacuum to obtain a solid residue containing squalene. As a control, solutions of standard squalene and lactic acid (30 ppm) were subjected to alkaline hydrolysis under the same experimental conditions to assess the eventual degradation of the analytes.

### 2.6. Quantification of Squalene through HPLC-DAD Analysis

HPLC-DAD method from [19] was adopted with slight modifications for the quantification of squalene. The dry residue containing squalene was solubilised in 1 mL of CH_3_CN using a sonicator. Subsequently, the sample was filtered using a 0.45-micron PTFE filter to be injected into HPLC (Agilent technology 1100 series, Santa Clara, CA, USA) coupled to a diode array detector (HPLC-DAD) method supported by ChemStation on an LC 3D system (Agilent, Santa Clara, CA, USA). The separation was performed isocratically on a RP18 reverse-phase column (Purospher, 5 µm 4.6 × 100 mm) using 100% CH_3_CN as the mobile phase at a flow rate of 1 µL/min. The injection volume was 1 μL for the squalene standard and 1 μL or 7 µL for the samples. The column temperature was set to 40 °C. Squalene was monitored and quantified at a maximum wavelength of 195 nm.

### 2.7. Quantification of Lactic Acid through HPLC-MS Analysis

The HPLC-MS method used for lactic acid quantification was modified from [20]. The aqueous phase of each sample was filtered with a 0.45-micron PTFE filter and injected into an HPMC Agilent Technologies 1290 Infinity with an Agilent Technologies 6420 triple quadrupole Triple Quad LC/MS. The source was drying gas (T = 300 °C, flow = 12 L/min, nebuliser = 55 psi). The column used was a Phenomenex SYNERGI (4 µm) Polar-RP 80A (150 × 4.60 mm). The mobile phase was formic acid 0.1% (A) with different percentages of methanol (B): from 0 to 5 min, 90% A and 10% B; from 5 to 10 min, 50% A and 50% B. The flow rate was set at 1 mL/min. The lactic acid standard and sample injection volume was 5 μL, and the column temperature was controlled at 30 °C. The acquisition mode was in single ion monitoring (SIM) for the 89 *m*/*z* ion, fragmentor 49. 

### 2.8. Validation of the HPLC Quantitative Methods

The validation of the analytical method involves several phases, such as evaluation of linearity, repeatability, calculation of the limit of detection (LOD) and of the limit of quantification (LOQ), and recovery [21]. For this purpose, standards at different concentrations were injected under the same experimental conditions used for the samples. 

### 2.9. Linearity

The linearity of the analytical method was evaluated by injecting into the instrument standards at increasing concentrations. For squalene, a 1000 ppm standard stock solution was prepared and then solutions at 5, 10, 50, 100, and 250 ppm, produced by diluting the stock solution with CH_3_CN, were then obtained. These samples were subsequently injected into HPLC, and the values of the areas were obtained by using an acquisition wavelength 195 nm. Regarding lactic acid, a stock solution was prepared using a 1000 ppm lactic acid standard stock solution and then preparing solutions at 0.1, 0.5, 1, 5, and 10 ppm by diluting with water. These samples were successively injected and the signal was recorded using ion acquisition at 89 *m*/*z*.

### 2.10. Limit of Detection (LOD) and Limit of Quantification (LOQ)

The sensitivity of the method was determined by assessing the limit of detection (LOD) and the limit of quantification (LOQ). The LOD was calculated from the concentrations at which the signal/noise ratio was equal to 3. The LOQ was calculated from the concentrations at which the signal/noise ratio was equal to 10.

### 2.11. Repeatability

The reproducibility of the method was evaluated by calculating the relative standard deviation (%RSD) between consecutive analyses (*n*  =  3) performed on the same day (intraday repeatability) and over three consecutive days (interday repeatability). For the analysis, a solution of the standard squalene at 10 ppm and a solution of standard lactic acid at 1 ppm were injected. 

### 2.12. Recovery

Recovery (%) was calculated from the ratio of the analyte area in the fortified sample to the analyte area in the unfortified sample. For squalene, the recovery tests were carried out at the concentration levels of 15 ppm and 50 ppm, while for lactic acid, they were conducted at the concentration levels of 5 ppm and 10 ppm.

## 3. Results

### 3.1. Evaluation of the Bulb Sheaths Damage through Polarised Light Microscopy

A preliminary evaluation of the morphological condition of the hair bulbs, before performing the extraction procedure, was carried out via observation with polarised optical microscopy. Such analysis further confirmed the classification of the samples into the groups of cases and controls, in addition to the medical anamnesis of the patient and trichoscopy. All samples grouped as cases (one bulb from 31 subjects for squalene and one bulb from 21 subjects for lactic acid) showed evident damage to the bulb sheaths. All samples grouped as controls did not show any evident damage to the bulb sheaths during microscopy examination. 

Figure 1 reports two pictures collected from samples as references. Specifically, Figure 1A shows a bulb in the anagen phase, which has intact sheaths (both inner and external). It appears without any remarkable imperfection from a physio-morphological point of view; therefore, all samples with the same appearance were used as controls. On the contrary, the two bulbs shown in Figure 1B clearly display some morphological imperfections with inner and external sheaths damaged, probably due to the presence of metabolic wastes such as squalene or lactic acid. All samples displaying similar features were confirmed as cases.

### 3.2. Selection of the Extraction Method

Different methods are reported in the literature for the extraction of squalene, both from hair or other matrices [16,19,22,23]. In order to establish the optimal extraction condition for hair bulbs, some preliminary trials were performed. Specifically, a methodology involving the use of a 2:1 *v*/*v* mixture of methanol/chloroform with subsequent sonication, evaporation of the solvents, and recovery of the extract with chloroform and methanol did not achieve the extraction of squalene from bulbs, probably since bulbs did not disintegrate in the extraction medium during the process [19]. Therefore, a method based on the alkaline hydrolysis of the hair bulb was applied, followed by extraction with n-hexane, subsequent evaporation of the solvent, and solubilisation of the extracted residue with acetonitrile [16]. This method provided a quantitative extraction of squalene from hair bulbs and was selected for the investigation. The number of consecutive extractions with n-hexane was also optimised by determining the amount of squalene extracted for each step. After five consecutive extractions, the amount of squalene was below the limit of determination; therefore, five was selected as the best condition. Additionally, since no extraction methods from hair are reported in the literature for lactic acid, it was assumed to be retained in the aqueous phase after the hydrolysis process due to its predominant hydrophilic characteristic. The extraction procedure at alkaline conditions did not affect the chemical stability of squalene or acid lactic. 

### 3.3. Validation of the Analytical Method

The HPLC-MS/MS method was validated in terms of linearity, limit of detection (LOD), limit of quantification (LOQ), and repeatability. The values for these parameters are reported in Table 1. The chromatograms of the squalene and lactic acid standards at 10 ppm from HPLC-DAD and HPLC-MS, respectively, are shown in Figure 2. According to the method used, squalene has a retention time of 8.3 min, and lactic acid has a retention time of 2.3 min. 

Linearity was optimal for both analytical methods employed for the quantification of squalene and lactic acid, being >0.998. Repeatability is expressed as relative standard deviation percentage (RSD%), which was in the range of 2.5–4.2 for the intraday precision and in the range of 3.1–3.6 for the interday precision. These values were slightly lower for the method used for lactic acid than that used for squalene determination, despite the concentration of standards used being different (10 ppm for squalene and 1 ppm for lactic acid). A slightly higher recovery (%) at both levels of concentration was determined for lactic acid extracted from bulbs rather than for squalene, being 93%/101% and 89.4%/92.7%, respectively. 

### 3.4. Quantification of Squalene and Lactic Acid in Hair Bulbs

Figure 3 and Figure 4 show the results obtained from the quantification of squalene and lactic acid from hair bulbs, classified as the group of “case” (i.e., subjects with damage to the sheaths observable through optical polarised microscopy) and as the group of the controls (i.e., subjects without apparently damage to the sheaths observable through optical polarised microscopy).

The relative frequency distribution (%) of the experimental data (Figure 3) is characterised by skewness values (0.2302 for cases and 0.8552 for controls) and kurtosis values (−0.3071 for cases and −0.3935 for controls), which are close to zero for the frequency distribution relative to the calculated squalene concentrations. For lactic acid samples, the skewness values were 1.682 for cases and 0.6774 for controls, and kurtosis values were 3.060 for cases and 0.2496 for controls. For both squalene and lactic acid, the data were normally distributed according to the Kolmogorov–Smirnov test, and they could be fitted with a Gaussian function, as shown in Figure 3.

The results of the quantification of squalene and lactic acid from bulbs in “cases” and ”controls” groups are presented as scatter plots (Figure 4). In these plots, the single experimental values are shown in the form of symbols; the mean and standard deviation are represented by the middle line and the upper and lower lines, respectively. With regard to squalene samples, minimum values of 2.58 µg/mL and 1.38 µg/mL, maximum values of 7.95 µg/mL and 4.44 µg/mL, and mean values of 5.17 µg/mL and 2.27 µg/mL were calculated for the cases and controls. For lactic acid samples, minimum values of 0.25 µg/mL and 0.24 µg/mL, maximum values of 3.25 µg/mL and 2.55 µg/mL, and mean values of 1.01 µg/mL and 1.06 µg/mL were calculated for the cases and controls. A statistical comparison was performed between the populations of the cases and controls using the unpaired two-tailed *t*-test with Welch’s correction test (Prism6; GraphPad Software; Boston, MA, USA). For squalene, it was found that the two populations (cases and controls) were statistically different (*p* < 0.001. On the contrary, no statistical differences were found between samples and controls for lactic acid (*p* = 0.7894). 

## 4. Discussion

Lipids are one of the main components of hair, both in the shaft (extended above the skin surface) and in the follicle (the portion beneath the skin). Hair lipids are generally classified as exogenous or endogenous according to their origin: sebaceous glands or matrix cells, respectively [24]. Ceramides, glycosylceramides, cholesterol sulphate, and 18-methyleicosanoic acid (18-MEA) are endogenous hair lipids. Triglycerides, wax esters, and squalene are exogenous hair lipids. Free fatty acids (FFAs) and cholesterol (CH) can have both exogenous and endogenous origins [25]. Few studies have investigated the lipid composition and content of human hair, both in physiological and pathological conditions [26,27,28]. Modifications of the hair lipid profile has been found in several dermatological and systemic diseases [29,30]. Specifically, an increase in the hair lipid content has been observed in patients affected by alopecia [29]. Most of these studies have focused on the analysis of the hair shaft [31], and only a limited number of papers have investigated the lipid composition of the hair follicle [24]. The scarce information on lipids in this analytical field may be related to the low amounts of analytes in these biological matrices and the difficulty of developing effective qualitative–quantitative methods [31]. In this work, HPLC was employed for the identification and quantification of squalene and lactic acid, other “metabolic wastes”, in hair bulbs. HPLC has already been used alone [19] or in combination with other analytical techniques, such as thin-layer chromatography flame ionization detection (TLC/FID) or gas chromatography coupled with mass spectrometry (GC/MS) [32], but its potential has not been fully explored. Therefore, HPLC with different detectors was applied for the quantification of squalene and lactic acid from hair in the follicles by cutting the collected hair from the scalp 3 mm from the end. Masukawa, et al. [32] reported an amount of squalene lower than 0.6 mg/g of hair, not varying much along the length of the hair itself. Another paper reports a mean percentage of 2.9% for squalene in the hair shaft with respect to total lipids [33]. Wu et al., have determined a squalene content in hair at different distances from the hair root of between 0.1% and 0.2% *w*/*w* using GC-MS. In contrast to squalene, lactic acid has never been quantified in hair samples. The rationale behind quantifying lactic acid is due to the fact that lactic acid is supposed to be accumulated in the sebaceous gland under continuous adrenaline stimulation. As a consequence, lactic acid is released in follicles and may have a role in the growth of hair. 

A limitation of this study is related to the quantification of squalene or lactic acid in a single bulb from each selected subject (31 subjects for squalene and 21 subjects for lactic acid). Each bulb was previously observed through optical microscopy to assess sheath damage. Therefore, it was possible to relate the content of squalene and lactic acid with the morphological alterations in the hair. However, biological variability in squalene and lactic acid content among bulbs extracted from the same subjects was not taken into account. 

## 5. Conclusions

This study focused on the extraction and characterisation of squalene and lactic acid as metabolic substances of the hair. Squalene and lactic acid were first extracted from hair bulbs and then successfully quantified via HPLC. Only squalene had a significantly larger amount quantified in subjects affected by alopecia with damage to hair sheaths. No statistically significant differences were found in terms of lactic acid concentration extracted from bulbs between the cases group and the controls, at least for the experimental conditions used. Therefore, squalene could be considered as a marker of hair pathological onsets for some alopecia conditions. Moreover, in addition to the assessment of the quantification method, this study allowed identification of a relationship between the microscopic images of damaged bulb sheaths and the actual amount of squalene in subjects with a defined clinical picture. Therefore, the extraction and analytical quantification of these substances can be considered as a first approach for setting up new pathogenic markers that can be exploited for the identification of pathological alopecia conditions in patients.

## Figures and Tables

**Figure 1 biomedicines-11-02493-f001:**
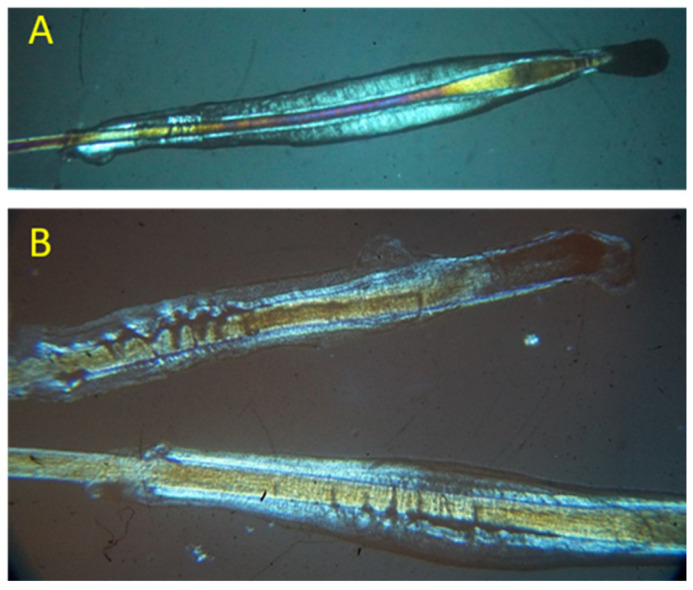
Images from optical polarised microscope of hair bulbs collected from the subjects involved in the study: hair bulb with intact inner and external sheaths (controls) (**A**) and hair bulbs with damaged inner and external sheaths (cases group) (**B**).

**Figure 2 biomedicines-11-02493-f002:**
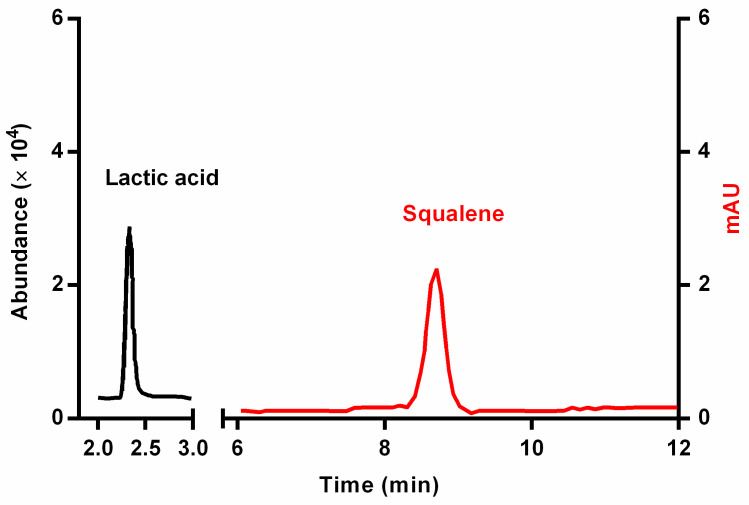
Chromatograms for the 10 ppm standard solutions of squalene and lactic acid recorded through HPLC-DAD and HPLC-MS, respectively.

**Figure 3 biomedicines-11-02493-f003:**
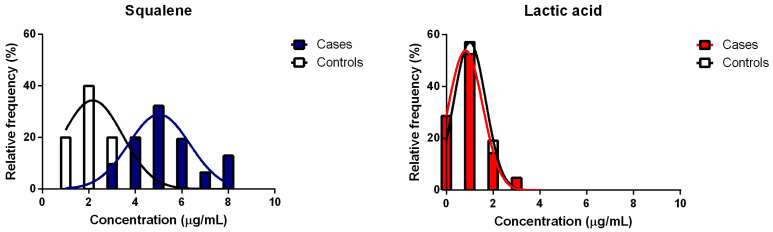
Relative frequency (%) distributions for the groups of cases and controls relative to the quantification of squalene and lactic acid from hair bulbs.

**Figure 4 biomedicines-11-02493-f004:**
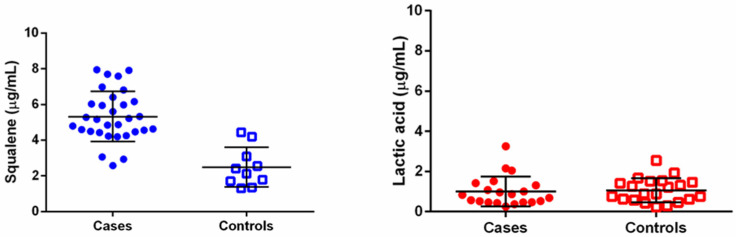
Scatter plot of cases and controls populations for the quantification of squalene and lactic acid in hair bulbs. Symbols represent the experimental data points (calculated concentration for squalene and lactic acid in each sample), while lines represent mean ± standard deviation. For squalene: samples (*n* = 31) and controls (*n* = 10). For lactic acid: samples (*n* = 21) and controls (*n* = 21).

**Table 1 biomedicines-11-02493-t001:** Validation parameters for the HPLC-DAD and HPLC–MS methods used for the quantification of squalene and lactic acid, respectively, in hair bulbs, and recovery (%) at two levels of concentration for the extraction procedure.

	Conc. Range (ppm)	R^2^	LOD (ppm)	LOQ (ppm)	Repeatability (RSD %, *n* = 3)		Recovery (%, *n* = 3)
					Intraday	Interday	
**Squalene**	5–250	0.9999	1	3	4.2	3.6	89.4/92.7
**Lactic acid**	0.1–10	0.9981	0.07	0.22	2.5	3.1	93.0/101

## Data Availability

The data presented in this study are available on request from the corresponding author.

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
