# Peer review of "Quantification of Squalene and Lactic Acid in Hair Bulbs with Damaged Sheaths: Are They Metabolic Wastes in Alopecia?"

_biomedicines, 2023, doi:10.3390/biomedicines11092493_

Round 1
Reviewer 1 Report
This is a quite well-written, compact study on squalene and lactic acid quantification in hair bulbs of people with alopecia. Although the number of objects involved in the investigation is not very large, the paper is a good preliminary study with a well-elaborated methodology. The introduction gives relevant background, the methods are clearly described, and the presentation of the results is good.
I have only a few minor questions/suggestions for the authors."
1) Authors used alkaline hydrolysis to extract analytes from the matrix. Have they checked whether lactic acid and squalene are stable under alkaline conditions? Add the explanation in the text.
2) Editorial errors: There are unnecessary capital letters in some places (see e.g. lines 62, 84, 86…). Line 123: lack of space „was1”; „ml” – it should be „mL”
3) The gradient program is unclearly described. „t = 3 min 10% B, t = 5 min 50%” – and what was composition of mobile phase between 3 and 5 min?
4) „The flow rate was set at 1 mL/min. (…) the column temperature was controlled at 30 °C.” and in next line: „The column temperature and flow rate were optimized to 25 °C and 0.2 mL/min, respectively.” Why were two different values given for temperature and flow rate? Which one represents the final conditions?
5) Editorial errors: There are unnecessary capital letters in some places (see e.g. lines 62, 84, 86…). Line 123: lack of space „was1”; „ml” – it should be „mL”
Author Response
This is a quite well-written, compact study on squalene and lactic acid quantification in hair bulbs of people with alopecia. Although the number of objects involved in the investigation is not very large, the paper is a good preliminary study with a well-elaborated methodology. The introduction gives relevant background, the methods are clearly described, and the presentation of the results is good.
The authors thank the reviewer for the positive comment on the manuscript.
I have only a few minor questions/suggestions for the authors."
1) Authors used alkaline hydrolysis to extract analytes from the matrix. Have they checked whether lactic acid and squalene are stable under alkaline conditions? Add the explanation in the text.
As a preliminary analysis, it was checked that squalene and lactic acid do not degrade under the alkaline conditions applied for the extraction by hydrolysis. This information has been added in the revised manuscript (paragraph 2.5 and paragraph 3.2).
2) Editorial errors: There are unnecessary capital letters in some places (see e.g. lines 62, 84, 86…). Line 123: lack of space „was1”; „ml” – it should be „mL
The editorial errors have been corrected.
3) The gradient program is unclearly described. „t = 3 min 10% B, t = 5 min 50%” – and what was composition of mobile phase between 3 and 5 min?
The gradient program has been rewritten for a better clarity.
4) „The flow rate was set at 1 mL/min. (…) the column temperature was controlled at 30 °C.” and in next line: „The column temperature and flow rate were optimized to 25 °C and 0.2 mL/min, respectively.” Why were two different values given for temperature and flow rate? Which one represents the final conditions?
The authors thank the reviewer for noticing this oversight. The sentence “The column temperature and flow rate were optimized to 25 °C and 0.2 mL/min, respectively” has been deleted from the manuscript since the real operating conditions were 30 °C and flow of 1 mL/min as reported above.
Reviewer 2 Report
The manuscript is interesting and produces a correct analysis of the results obtained. The experimental design is correct and the methods applied are well-managed. The evidence associated with the presence of squalene in a large part of the patients analyzed introduces an interesting point of view useful in the management of the alopecia in particular in the early stage.
For this aspect, I suggest accepting the manuscript in its present form.
Author Response
The manuscript is interesting and produces a correct analysis of the results obtained. The experimental design is correct and the methods applied are well-managed. The evidence associated with the presence of squalene in a large part of the patients analyzed introduces an interesting point of view useful in the management of the alopecia in particular in the early stage.
For this aspect, I suggest accepting the manuscript in its present form.
The authors thank the reviewer for the positive comment on the manuscript.
Reviewer 3 Report
XThe manuscript "Quantification of squalene and lactic acid in hair bulbs with damaged sheaths: are they metabolic wastes in alopecia condition?" fits the journal 's scope. The authors present their results on the extraction, identification and quantification of squalene and lactic acid from hair bulbs.
Although the results are clearly presented and the quality of presentation is sufficient, the authors should clarify some major issues.
1. Please add clarification regarding the novelty of the work. Lactic acid and squalene were previously reported to be involved in alopecia condition, in the literature being available article and patents.
2. The aim of the study seems to be the development/optimization of the quantification methods of squalene and lactic acid, and the application of these methods on different samples. Thus, the title does not reflect the content of the work.
3. Althought the authors described the selection criteria, more data are needed (please see below). The aim of the work and the research design are sufficiently presented.
4. Section 2.4 - please add information regarding the selection site, number of part
Section 2.6, 2.7 - please add the missing reference(s). For both methods the references should be indicated; the optimization steps of the identification and quantification methods should be clearly indicated.
5. Validation of analytical methods - please indicate the references/guidelines followed
Linearity - please add justification for using only five concentrations, and not six.
Please add justification for assessing the recovery at only one level of concentration.
Table 1 please correct LOD and LOQ.
Please explain the rationale for choosing the range of concentrations of squalene and lactic acid. The minimum values of determined samples are under the LOQ for squalene (if the authors switched the values of LOD and LOQ by mistake), and very close to LOQ for lactic acid, in both cases being out of the linearity range.
Formatting and typing/English errors
Lines 123 -please correct the errors
Author Response
The manuscript "Quantification of squalene and lactic acid in hair bulbs with damaged sheaths: are they metabolic wastes in alopecia condition?" fits the journal 's scope. The authors present their results on the extraction, identification and quantification of squalene and lactic acid from hair bulbs.
Although the results are clearly presented and the quality of presentation is sufficient, the authors should clarify some major issues.
1. Please add clarification regarding the novelty of the work. Lactic acid and squalene were previously reported to be involved in alopecia condition, in the literature being available article and patents.
The novelty of the work relies on the assessment and the analytical quantification of squalene and lactic acid in the hair bulbs of subjects having an alopecia condition. The involvement of these compounds as metabolic “wastes” able to damage hair in alopecia conditions has been only supposed in previous studies available in the literature but never demonstrated scientifically through their analytical quantification. This aspect has been better specified in the introduction section.
2. The aim of the study seems to be the development/optimization of the quantification methods of squalene and lactic acid, and the application of these methods on different samples. Thus, the title does not reflect the content of the work.
As stated at the end of the introduction, the aim of the work was the extraction and analytical quantification of squalene and lactic acid from hair bulbs of subjects selected on the basis of their clinical history of alopecia and after a preliminary evaluation of the bulbs through polarized light optical microscopy. Therefore, the study is not aimed to develop and optimize a new analytical method for the quantification of squalene and lactic acid on different samples, since these methods are available in the literature and also cited in the manuscript (References 19 and References 22-23). The method reported in the references 19 has been adopted with some modifications for the extraction of both squalene and lactic acid in this study. These metabolites were extracted and quantified from hair bulbs to correlate the results with those collected from the observation of the same bulbs through polarised optical microscopy. All the bulbs identified as the “samples” group and collected from subjects affected by alopecia showed a clear sheaths damage by polarised optical microscopy. The quantification of lactic acid and squalene was performed to assess whether these compounds can be involved in the morphological features of bulbs observed in alopecia conditions. According to this premises, the title reflects the scope and the content of the study presented in the manuscript. However, to improve the scientific rightness of the manuscript, the word “optimization” was deleted from line 75 and the aim of the work at the end of the introduction has been slightly modified. Moreover, the references regarding the analytical HPLC method adopted has been added in the method section (paragraphs 2.6 and 2.7).
3. Although the authors described the selection criteria, more data are needed (please see below). The aim of the work and the research design are sufficiently presented. 4. Section 2.4 - please add information regarding the selection site, number of part
The site for sample collection on subjects was added in paragraph 2.2. The number of the subjects (cases and controls) involved in the study both for squalene and lactic acid is clearly indicated in the paragraphs 3.1 and 3.4.
Section 2.6, 2.7 - please add the missing reference(s). For both methods the references should be indicated; the optimization steps of the identification and quantification methods should be clearly indicated.
The references regarding the analytical HPLC methods for both squalene and lactic acid have been added in paragraphs 2.6 and 2.7.
5. Validation of analytical methods - please indicate the references/guidelines followed
A reference has been added in paragraph 2.8 regarding the guidelines used for the validation of the method.
Linearity - please add justification for using only five concentrations, and not six.
Please explain the rationale for choosing the range of concentrations of squalene and lactic acid. The minimum values of determined samples are under the LOQ for squalene (if the authors switched the values of LOD and LOQ by mistake), and very close to LOQ for lactic acid, in both cases being out of the linearity range.
The authors agree with the reviewer that few samples of squalene “cases” and all samples for squalene “controls” are apparently out of the linearity range according to the concentrations used for the calibration curve (5-250 ppm) and some of these samples are also apparently below the LOQ. However, HPLC analysis for these samples was also repeated by injecting a larger volume (7 µL instead of 1 µL) of samples to detect a concentration that is both in the linearity range and above the LOQ. This information has been added in the revised manuscript (paragraph 2.6). On the contrary, all samples for lactic acid were in the range of linearity (0.1-10 ppm) and above the LOQ of 0.22.
Please add justification for assessing the recovery at only one level of concentration.
The authors thank the reviewer to have raised this point. Actually, the recovery was performed at two levels of concentration, but in the previous version only one level of concentration was reported. The revised version of the manuscript (Table 1, paragraph 2.8 and paragraph 3.3) has been updated reporting two levels of concentration for the recovery of squalene (15 and 50 ppm) and lactic acid (5 and 10 ppm).
Table 1 please correct LOD and LOQ.
The authors thank the reviewer for highlighting this oversight. Table 1 has been corrected.
Formatting and typing/English errors
Done
Lines 123 -please correct the errors
Done
Round 2
Reviewer 3 Report
The authors have addressed the raised points, incorporating necessary corrections or providing appropriate justifications. In its current form, the manuscript is suited for publication.